# Novel Pyridine Bioisostere of Cabozantinib as a Potent *c*-Met Kinase Inhibitor: Synthesis and Anti-Tumor Activity against Hepatocellular Carcinoma

**DOI:** 10.3390/ijms22189685

**Published:** 2021-09-07

**Authors:** Ujjwala Karmacharya, Diwakar Guragain, Prakash Chaudhary, Jun-Goo Jee, Jung-Ae Kim, Byeong-Seon Jeong

**Affiliations:** 1College of Pharmacy, Yeungnam University, Gyeongsan 38541, Korea; Karmacharya@ynu.ac.kr (U.K.); diwakarguragain@ynu.ac.kr (D.G.); prakash@ynu.ac.kr (P.C.); 2College of Pharmacy, Kyungpook National University, Daegu 41566, Korea; jjee@knu.ac.kr

**Keywords:** mesenchymal–epithelial transition factor (*c*-Met), bioisosteric replacement, anti-proliferative activity, tumor selectivity, hepatocellular carcinoma, anti-tumor efficacy

## Abstract

Two novel bioisosteres of cabozantinib, **3** and **4**, were designed and synthesized. The benzene ring in the center of the cabozantinib structure was replaced by trimethylpyridine (**3**) and pyridine (**4**), respectively. Surprisingly, the two compounds showed extremely contrasting mesenchymal–epithelial transition factor (*c*-Met) inhibitory activities at 1 μM concentration (4% inhibition of **3** vs. 94% inhibition of **4**). The IC_50_ value of compound **4** was 4.9 nM, similar to that of cabozantinib (5.4 nM). A ligand-based docking study suggested that **4** includes the preferred conformation for the binding to *c*-Met in the conformational ensemble, but **3** does not. The anti-proliferative activity of compound **4** against hepatocellular carcinoma (Hep3B and Huh7) and non-small-cell lung cancer (A549 and H1299) cell lines was better than that of cabozantinib, whereas **3** did not show a significant anti-proliferative activity. Moreover, the tumor selectivity of compound **4** toward hepatocellular carcinoma cell lines was higher than that of cabozantinib. In the xenograft *chick* tumor model, compound **4** inhibited Hep3B tumor growth to a much greater extent than cabozantinib. The present study suggests that compound **4** may be a good therapeutic candidate against hepatocellular carcinoma.

## 1. Introduction

The mesenchymal–epithelial transition factor (*c*-Met) is a receptor tyrosine kinase, also called hepatocyte growth factor receptor (HGFR) [1,2,3]. HGF or scatter factor (SF), which is the only high-affinity ligand for *c*-Met, is produced by cells of mesenchymal origin and serves as the most potent hepatocyte mitogen [1,4]. *c*-Met activation plays a pivotal role in cell survival, cell morphogenesis, proliferation, embryo development, and angiogenesis [3,5]. In organ injury, *c*-Met also acts to stimulate tissue repair and regeneration [6,7]. An abnormal activation of *c*-Met by transcriptional upregulation and gene mutation is observed in several types of solid tumors [8]. Such an aberrant *c*-Met signal acts as a primary and secondary driver mutation to induce tumor growth and acquired resistance to targeted therapy [9,10,11,12]. In patients with hepatocellular carcinoma (HCC) and non-small-cell lung cancer (NSCLC), aberrant *c*-Met alternations like point mutations, amplification, and fusion are associated with drug resistance, aggressiveness, and poor prognosis of the patients [9,13,14,15,16]. Based on the functionality of *c*-Met and the clinical efficacy of monoclonal antibody and small-molecule inhibitors, which is meaningful mainly in patients with NSCLC and HCC [5], *c*-Met has emerged as one of the promising targets for the development of anti-cancer therapeutics against, mainly, HCC and NSCLC.

The *c*-MET receptor is composed of an α and a β chain linked by a disulfide bond. The Intracellular kinase domain in the β chain contains catalytic tyrosine residues, Y1234 and Y1235, which positively regulate the enzyme activity, whereas tyrosine residue Y1003 in the juxtamembrane domain negatively regulates the enzyme [17]. Although Y1349 and Y1356 in the *C*-terminal tail multifunctional docking site of the β chain are phosphorylated at a later time point after the phosphorylation of Y1234 and Y1235, Y1349 and Y1356 recruit signal transducers and adaptors, and their activity has been suggested as a predictor of metastasis and patient survival [18]. Three types of *c*-Met inhibitors have been developed so far: (i) selective *c*-Met tyrosine kinase inhibitor, (ii) multi-tyrosine kinase inhibitors, and (iii) monoclonal antibody against c-Met [19,20]. Over the last decade, a few compounds have been approved by the US Food and Drug Administration (FDA) to use in patients with NSCLCs (crizotinib, 2011) [21], medullary thyroid cancer (cabozantinib, 2012) [22], and kidney cancer (cabozantinib, 2016) [23].

Small molecule *c*-Met kinase inhibitors targeting the catalytic tyrosine kinase residues have been classified into type I and type II based on the binding modes with *c*-Met kinase [24,25]. Type I inhibitors represented by crizotinib generally have a U-shaped conformation for an effective binding to the activation loop of *c*-Met. Type II inhibitors are known to bind to *c*-Met with an extended conformation stretching from the ATP-binding site to the deep hydrophobic region and to have urea or urea-like common structures. Bioisosterism is a tool for the rational modification of lead compounds searching for analogues aiming at the improvement of undesirable characteristics [26,27]. A bioisosteric replacement exchanges a group of atoms of a lead compound with a similar group of compounds expecting better affinity, efficacy, druggability, and/or lower toxicity than the lead compound [28]. Diverse and complementary results are often retrieved from the application of this strategy, and thus it is one of the reliable methods for the rational optimization of lead compounds [29].

With the application of the bioisosteric replacement concept, we recently reported a novel hybrid compound, **1**, with an improved safety window compared to the parent molecule, sunitinib, which is a multi-tyrosine kinase inhibitor currently used as an anti-cancer drug (Figure 1) [30]. 5-Fluoroindolin-2-one in the structure of sunitinib was replaced with a pyridine-containing 5-hydroxy-4,6-dimethyl-1,3-dihydro-2*H*-pyrrolo[2,3-*b*]pyridin-2-one (**2**), which was prepared from a natural product, pyridoxine HCl. As in this case, pyridine moieties are frequently employed in the bioisosteric replacement of the corresponding benzene counterparts of parent molecules because of their properties, such as their ability to form hydrogen bonds, their basicity, and their solubility in water. This compound **1** showed a comparable anti-cancer activity against several cancer cell lines and was less toxic to normal cell lines when compared to sunitinib. It has a higher safety window than sunitinib and may act as an advantageous factor in the drug development field. This promising result showed the practicability of this strategy for new drug design and synthesis.

As an extension of our efforts for the discovery of new, promising, and biologically active compounds by bioisosterism, in this study we chose a representative type II *c*-Met tyrosine kinase inhibitor, cabozantinib, as a target molecule. We designed two novel compounds, **3** and **4**, by replacing the centered benzene ring in cabozantinib structure by pyridoxine-derived 2,4,5-trimethylpyridine (for **3**) and non-substituted pyridine (for **4**), respectively. We aimed to investigate how this structural change affects *c*-Met inhibitory activity, and to elucidate a relationship between structure and activity using a ligand-based docking study. Finally, the anti-tumor effect of the bioisostere was confirmed in a xenograft tumor model.

## 2. Results

### 2.1. Synthesis of Cabozantinib Bioisosteres ***3*** and ***4***

The syntheses of the new compounds, **3** and **4**, are shown in Figure 1. First, the synthesis of the key synthetic intermediate, 2,4,5-trimethylpyridin-3-ol (**9**), for the trimethyl-substituted pyridine ring-containing compound (**3**) was started from commercially available pyridoxine HCl (vitamin B_6_) [31]. A dichlorination of two benzylic alcohols of pyridoxine using thionyl chloride in the presence of a catalytic amount of *N*,*N*-dimethylformamide was done, and this reaction was followed by the reductive cleavage of the obtained product in refluxing acetic acid to get compound 2,4,5-trimethylpyridin-3-ol (**5**). Further bromination with 1,3-dibromo-5,5-dimethylhydantoin was carried out, and the product obtained was treated with benzyl chloride to protect the phenolic-hydroxy-group-affording compound (**6**). The bromo compound, **6**, was then aminated under Buchwald–Hartwig amination reaction conditions with benzophenone imine to give compound (**7**). The imine group of (**7**) was then methanolized, affording amine compound (**8**), which on debenzylation gave the key ring compound (**9**).

Next, for the synthesis of non-substituted pyridine analogue 4, 6-aminopyridin-3-ol (**12**) was prepared from commercially available pyridin-3-ol (**10**) [32]. A nitrogen atom was introduced at the *para*-position of the hydroxyl group in 10 via an azo-coupling reaction with a *p*-nitrophenol-affording compound, **11**. Then, the cleavage of the azo bond under hydrogenolysis conditions yielded the amino compound **12**. These two key rings, **9** and **12**, were then coupled with commercially available 4-chloro-6,7-dimethoxyquinoline (**15**) for ethereal bond formation to afford the **16** and **17** compounds. These nucleophilic aromatic substitution reactions were done under either microwave irradiation in the presence of potassium *tert*-butoxide in *N*,*N*-dimethylformamide (for **16**) or conventional reaction conditions with sodium hydride in dimethyl sulfoxide (for **17**). Finally, a 1-((4-fluorophenyl)carbamoyl)cyclopropane-1-carbonyl moiety was introduced into the amino group of compounds **16** and **17** using the corresponding acid chloride, **14**, which was prepared from commercially available carboxylic acid, **13**, to afford the final products—**3** and **4**, respectively.

### 2.2. Comparison of c-Met Kinase Activity and Ligand-Based Docking

The inhibitory activities of compounds **3** and **4**, the novel bioisosteres of cabozantinib against *c*-Met kinase, were dramatically different. At a fixed concentration (1 μM), compound **3**, with the trimethylpyridine ring center, inhibited *c*-Met only by 3.8%, whereas the unsubstituted pyridine-centered compound, **4**, showed 94.3% of *c*-Met inhibition. The half maximal inhibitory concentrations (IC_50_) of cabozantinib and its pyridine-bioisostere **4** were 5.4 and 4.9 nM, respectively (Table 1).

To explain the experimental results at the molecular level, we performed the ligand-based docking study. No crystal structure of cabozantinib with kinase has been reported. Of the 63 inhibitors found in the crystal complex with *c*-Met, the one showing the highest TanimotoCombo score to cabozantinib and the unsubstituted pyridine-centered compound, **4**, was compound **18** in Figure 2, an analog of altiratinib, with the values of 1.357 and 1.306, respectively. The PDB ID containing the molecule is 5DG5, where the ligand ID is 5B4. The aligned cabozantinib and compound **4** structures fitted into the coordinates of 5DG5 support the close similarity of the intermolecular interaction in the cabozantinib, compound **4**, and 5B4. In particular, the models preserved the intermolecular hydrogen bonds to Met-1160 and Asp-1222. On the other hand, the TanimotoCombo score between the trimethylpyridine-centered compound, **3**, and 5B4 was 0.983. The pair of compound **3** and the ligand of 3EFK, compound **19** (Ligand ID: MT4), showed the highest TanimotoCombo score, 1.306. MT4 has a different chemical scaffold in the part corresponding to (4-fluorophenyl)cyclopropane-1,1-dicarboxamide of 5B4. Moreover, the fitted complex structure of compound **4** into 3EFK showed the many intermolecular clashes that potentially hinder binding to compound **3**. Our *in-silico* study suggests that the ligand-only structures can sufficiently reveal the differences in the local geometries of cabozantinib, compound **3**, and compound **4**. It should be noted that the ensemble of well-sampled ligand conformers can include the conformer found in the complex structure [33,34]. Nevertheless, the ensemble of compound **3** contained a small quantity of the relevant conformer, which is consistent with the experimental data.

### 2.3. Anti-Proliferative Activities of Compounds ***3*** and ***4***

The new *c*-Met inhibitor, compound **4**, was examined for its anti-proliferative activity against the HCC and NSCLC cancer cell lines. Compound **4** was observed to have a more potent anti-proliferative activity than cabozantinib against HCC cancer cell lines (Hep3B and Huh7). By contrast, compound **3** did not show a significant anti-proliferative activity. The IC_50_ values of cabozantinib and compound **4** against Hep3B cell proliferation were 15.2 and 2.1 μM, respectively, and 9.1 and 6.2 μM against the Huh7 cell line, respectively (Figure 3 and Table 2).

In the NSCLC cell lines, the anti-proliferative activity of compound **4** was better than that of cabozantinib. The IC_50_ values of compound **4** against A549 and H1299 cell proliferation were 12.4 and 8.9 μM, respectively, and those of cabozantinib were 37.3 and 16.1 μM, respectively. In addition, the cytotoxicity of compound **4** against a normal human pancreatic duct epithelial cell line, H6c7, was a little bit higher than that of cabozantinib. However, the IC_50_ values of compound **3** against proliferation and cytotoxicity in all five cell lines were over 100 μM.

We also examined the tumor selectivity (TS) of compound **4** and compared it to that of cabozantinib by calculating TS as the ratio of the IC_50_ values for each compound on the cancer cells versus H6c7 normal epithelial cells (Table 3). The TS analysis showed that compound **4** has a higher TS toward HCC cell lines than NSCLC cell lines, and the HCC selectivity of compound **4** was much higher compared to that of cabozantinib.

### 2.4. Suppression of c-MET Expression in HCC and NSCLC by Compound ***4***

In order to reveal the action mechanism by which compound **4** induces a much better inhibitory effect on cancer cell proliferation than cabozantinib, we examined whether two compounds possess differential activity to suppress *c*-MET expression. At basal condition, the *c*-MET protein expression level was greater in A549 cells than in the other three cell lines, H1299, Hep3B, and Huh7 (Figure 4A). However, the phosphorylation levels of catalytic site tyrosine residues, Y1230, Y1234, and Y1235, were lowest in A549 cells, followed by H1299, Hep3B, and Huh7 cells (Figure 4A). On the other hand, Y1349 phosphorylation at basal condition was much higher in NSCLC than in HCC cell lines. The treatment of the cancer cells with cabozantinib and compound **4** at the same concentration significantly reduced the expression of *c*-MET and its phosphorylation in a time-dependent manner, and such change by compound **4** was much greater than by cabozantinib (Figure 4B). In a comparison between Hep3B and Huh7 cells, the inhibitory effect of compound **4** on *c*-MET expression at 48 h was greater in Hep3B than in Huh7 cells. In the case of NSCLC cell lines, the *c*-MET decrease in H1299 cells was greater than that in A549 cells.

### 2.5. Comparison of Apoptosis-Inducing Activity of Cabozantinib and Compound ***4***

Because cabozantinib induces apoptosis in colon cancer cells independent of p53 activity [36], the apoptosis-inducing activity of cabozantinib and compound **4** in Hep3B and A549 cells was also evaluated by fluorescence-activated cell sorting (FACS). As shown in Figure 5, the treatment of the cells with 10 μM of cabozantinib and compound **4** for 48 h induced apoptosis. Cabozantinib increased mostly early apoptotic cells, whereas compound **4** increased both the early and late-stage apoptotic/dead cell population in both Hep3B and A549 cells. In addition, the apoptosis induction by compound **4** was much greater in Hep3B than in A549 cells.

### 2.6. Anti-Tumor Activity of Compound ***4*** on a Chick Chorioallantoic Membrane (CAM) Tumor Model Implanted with Hep3B Cells

Next, the in vivo efficacy of compound **4** on the growth of Hep3B cells xenografted onto CAM was compared with that of cabozantinib. In the vehicle-treated control group, the implanted Hep3B cells developed a tumor mass and tumor-induced angiogenesis (Figure 6). The inhibitory effect of compound **4** on the tumor weight was greater than that of cabozantinib. Similarly, the tumor-induced angiogenesis was also significantly blocked by compound **4** and cabozantinib, and compound **4** showed a greater inhibition than cabozantinib.

## 3. Discussion and Conclusions

Two novel bioisosteres (compounds **3** and **4**) of cabozantinib were designed and synthesized by replacing the centered benzene ring of cabozantinib with trimethylpyridine (for **3**) and unsubstituted pyridine (for **4**), respectively. A significant difference in the inhibitory activity of the two compounds against *c*-Met kinase was observed: The trimethylpyridine analog **3** was not effective, whereas the unsubstituted pyridine analog **4** showed a potent inhibition comparable to cabozantinib. A ligand-based docking study suggested that such a differential activity of compound **3** may be due to a conformational hindrance caused by trimethyl groups in the pyridine ring in binding to *c*-Met. Considering the close similarity of cabozantinib and compound **4** in the chemical structure and the binding mode, the difference in their cellular activities likely arises from the other non-target effects.

Previously, it has been reported that the *c*-MET expression level in A549 cells is medium [11] or low [37] among NSCLC cell lines, and that it is functionally active. In the present study, although the basal-level *c*-MET expression was found to be higher in A549 cells than in H1299, Hep3B, and Huh7 cells, the activity (phosphorylation) status of *c*-MET catalytic kinase residues (Y1230, Y1234, and Y1235) in the presence of serum was not different from that of the other three cell lines, H1299, Hep3B, and Huh7 cells. In other words, it is possible that the *c*-MET kinase of the cancer cells with p53 null (Hep3B) or mutated (H1299 and Huh7) may be a little more active than A549 cells with the p53 wild type, based on the previous report that there is an inverse relationship between p53 status in cancer cells and HGF/*c*-MET signaling activity [38]. On the other hand, A549 and H1299 cells at basal condition showed a high level of phosphorylation in Y1349, which is the portion of the SH2 recognition motif recruiting signaling molecules [39]. The results implicate that, compared to HCC cell lines, NSCLC cells in the presence of serum could be in transactivation or crosstalk with other molecules [40], such as EGFR, overexpressed in NSCLC [41].

Despite the basal *c*-MET expression status, the long-term treatment of cancer cells with cabozantinib and compound **4** up to 48 h showed better anti-proliferative activity against Hep3B and Huh7 than A549 and H1299 cells. This result is consistent with the previous report that cabozantinib is more effective in cancer cells with mutated p53 [42]. In addition, the anti-proliferative activity of compound **4** was better than that of cabozantinib, and such a differential activity was similarly observed in the time-dependent down-regulation of *c*-MET expression. The downregulation of *c*-MET expression proceeds in two ways. One is ubiquitin-ligase-mediated degradation [43], and another is proteolytic cleavage by caspases or sequential cleavage by metalloproteases and γ-secretase [44]. Moreover, *MET* gene expression itself is controlled by various levels of regulatory mechanisms—not only transcriptional, but also epigenetic and post-transcriptional regulations [45]. Although the exact mechanism by which cabozantinib and compound **4** induce *c*-MET downregulation is not revealed in the present study, there was a correlation between anti-proliferative activity and the reduction of the *c*-MET expression level in cancer cells. Furthermore, compound **4** was a stronger *c*-MET inhibitor and anti-proliferative agent against HCC cell lines than cabozantinib. In the opposite direction, compound **3** also confirmed the correlation between *c*-MET inhibitory activity and anti-proliferative action against cancer cells. That is, compound **3**, having a very weak *c*-MET inhibitory ability, had a very low anti-proliferative activity against cancer cells. In addition, like cabozantinib, which induces cancer cell apoptosis regardless of the p53 status [36], compound **4** induced early and late-stage apoptosis with greater intensity than cabozantinib in both Hep3B and A549 cells. As predicted from the in vitro activities, the in vivo CAM tumor model capable of measuring tumor growth along with tumor-induced angiogenesis verified the stronger anti-tumor activity of compound **4**. Most importantly, the stronger *c*-MET inhibitor activity of compound **4** may be applicable to drug-resistant HCC cells, in which HGF expression is upregulated and the subsequent autocrine HGF/*c*-Met signaling is activated [46].

In conclusion, compound **4**, which possess a dual mode of action—the inhibition of both the activity and expression of *c*-MET—showed a much stronger HCC selectivity and anti-tumor activity than cabozantinib. The results suggest that compound **4** may be one of the promising therapeutic options against HCC.

## 4. Materials and Methods

### 4.1. Synthesis

#### 4.1.1. General

Unless noted otherwise, materials were purchased from commercial suppliers (Merck, Kenilworth, NJ, USA; ThermoFisher Scientific, Waltham, MA, USA; Tokyo Chemical Industry, Tokyo, Japan; Daejung Chemicals & Metals Co., Siheung, Korea, etc.) and used without further purification. Air or moisture-sensitive reactions were carried out under an inert gas atmosphere. The reaction progress was monitored by thin-layer-chromatography (TLC) using silica gel F_254_ plates. The products were purified by flash column chromatography using silica gel 60 (70–230 mesh) or using the Biotage “Isolera One” system with the indicated solvents. The melting points were determined using a Fisher–Johns melting point apparatus and were not corrected. Low-resolution mass spectra (LRMS) were obtained using an Advion Expression CMS and recorded in a positive ion mode with an electrospray (ESI) source. NMR spectra were obtained using a Bruker-250 spectrometer (250 MHz for ^1^H NMR and 62.5 MHz for ^13^C NMR) and a Bruker Avance Neo 400 spectrometer (400 MHz for ^1^H NMR and 100 MHz for ^13^C NMR). Chemical shifts (δ) were expressed in ppm using a solvent as an internal standard and the coupling constant (*J*) in hertz. HPLC analyses were performed on a system consisting of an LC-20AD pump, a CBM-20A communication bus module, an SPD-20A UV-visible detector, and a DGU-20A5 degasser from Shimadzu Corporation (Kyoto, Japan). A Phenomenex Luna^®^ C18 column (250 × 4.6 mm 5.0 μm) was used with a gradient solvent system consisting of acetonitrile and water (from 10 to 100% of acetonitrile over 15 min., then 100% of acetonitrile for 10 min.) at a flow rate of 1.0 mL/min at 254 nm UV detection. The purity of the compound was recorded as a percentage (%), and the retention time was given in min.

##### 6-[2-(4-Nitrophenyl)diazenyl)]-3-pyridinol (**11**) [CAS RN 35771-41-6]

*p*-Nitroaniline (800 mg, 5.79 mmol) was added slowly to a cooled 6 M hydrochloric acid in an iced bath. To this mixture, we added a solution of sodium nitrite (399 mg, 5.79 mmol) in water (2.5 mL) and stirred for 5 min. The mixture was added dropwise to a cooled mixture of pyridin-3-ol (**10**, 500 mg, 5.26 mmol) in water (2.5 mL), which was placed in an iced bath. During this addition, the pH of the solution was maintained at 8 with 6 M potassium hydroxide. After completion of the addition, the solution was brought to room temperature and was stirred for 90 min. The reaction mixture was then chilled again in an iced bath, and the pH was adjusted to 3 with hydrochloric acid and stirred for an additional 30 min. The precipitates in the mixture were collected by filtration, and the filter cake was successively washed with iced water, hexanes, ethyl ether, and dichloromethane and dried to afford 11 (1.08 g, 84%) as a bright orange solid. ^1^H NMR (DMSO-*d*_6_) δ 8.38–8.29 (m, 2H), 7.96–7.82 (m, 3H), 7.77 (d, *J* = 9.1 Hz, 1H), 6.77 (dd, *J* = 9.1, 2.9 Hz, 1H).

##### 6-Aminopyridin-3-ol (**12**) [CAS RN 55717-46-9]

A mixture of **11** (900 mg, 3.7 mmol), 10% palladium on activated carbon (37 mg) in methanol, was stirred at room temperature for 18 h under a hydrogen atmosphere. The mixture was filtered through a celite pad, and the filtrate was treated with a methanolic hydrochloride solution until the pH of the solution reached 3. It was then concentrated, and the residue was washed with hexanes and dichloromethane several times to afford 12 (412 mg, 76%) as a dark brown solid. ^1^H NMR (DMSO-*d*_6_) δ 8.60 (s, 1H), 7.50 (d, *J* = 3.0 Hz, 1H), 6.90 (dd, *J* = 8.7, 3.0 Hz, 1H), 6.34 (d, *J* = 8.7 Hz, 1H), 5.18 (s, 2H).

##### 5-((2,3-Dimethoxypyridin-4-yl)oxy)-3,4,6-trimethylpyridin-2-amine (**16**)

A mixture of **9** (270 mg, 1.7 mmol), potassium *tert*-butoxide (318 mg, 2.8) in *N*,*N*-dimethylformamide (15 mL) in a microwave reaction vial, was stirred for 30 min at room temperature. 4-Chloro-6,7-dimethoxyquinoline (**15**) (40 mg, 0.18 mmol) was added to it, and then the mixture was microwaved at 200 °C for 2 h at 200 W power. The mixture was concentrated, and the residue was diluted with ethyl acetate and water. The aqueous layer was extracted with ethyl acetate, and the combined organic solution was dried over anhydrous magnesium sulfate, filtered, and concentrated. The residue was purified by silica gel column chromatography (CH_2_Cl_2_:MeOH = 20:1) to afford 16 (400 mg, 67%) as a brown solid. R_f_ 0.30 (CH_2_Cl_2_:MeOH = 9:1); m.p. 103 °C; MS *m/z* 340.4 [M + H]^+^; ^1^H NMR (CDCl_3_) δ 8.43 (d, *J* = 5.3 Hz, 1H), 7.63 (s, 1H), 7.43 (s, 1H), 6.19 (d, *J* = 5.2 Hz, 1H), 4.44 (s, 2H), 4.06 (d, *J* = 7.7 Hz, 6H), 2.19 (s, *J* = 3.8 Hz, 3H), 2.09 (s, 3H), 2.03 (s, 3H) (see Appendix A); ^13^C NMR (CDCl_3_) δ 159.81, 153.96, 152.92, 149.58, 148.92, 146.67, 145.85, 139.93, 139.86, 115.37, 114.01, 107.87, 101.37, 99.42, 56.24, 56.18, 18.49, 13.11, 12.75 (see Appendix A); HPLC retention time 13.3 min, purity 99.3% (see Appendix A).

##### 5-((6,7-Dimethoxyquinolin-4-yl)oxy)pyridin-2-amine (**17**)

**12** (130 mg, 1.18 mmol) was added to a suspension of sodium hydride (28 mg, 1.18 mmol) in dimethyl sulfoxide (5 mL), and it was stirred for 10 min at room temperature. To this mixture, **15** (264 mg, 1.18 mmol) was added, and it was stirred at 90 °C for 1 h. The solvent was evaporated, and the residue was diluted with dichloromethane and water. The aqueous layer was extracted with dichloromethane, and the combined organic solution was dried over anhydrous magnesium sulfate, filtered, and concentrated. The residue was purified by silica gel column chromatography (CH_2_Cl_2_:MeOH = 20:1) to give 17 (78 mg, 22%) as a brown solid. R_f_ 0.20 (CH_2_Cl_2_:MeOH = 12:1); m.p. 180 °C; MS *m/z* 298.4 [M + H]^+^; ^1^H NMR (CD_3_OD) δ 8.29 (d, *J* = 5.5 Hz, 1H), 7.78 (d, *J* = 2.9 Hz, 1H), 7.46 (s, 1H), 7.30 (dd, *J* = 9.0, 2.9 Hz, 1H), 7.17 (s, 1H), 6.60 (d, *J* = 8.9 Hz, 1H), 6.39 (d, *J* = 5.5 Hz, 1H), 3.87 (d, *J* = 2.7 Hz, 6H) (see Appendix A); ^13^C NMR (CD_3_OD) δ 161.90, 157.83, 153.57, 149.94, 147.92, 145.56, 142.14, 139.50, 131.98, 115.65, 109.80, 105.73, 102.03, 99.15, 55.10, 55.08 (2C) (see Appendix A); HPLC retention time 12.1 min, purity 98.1% (see Appendix A).

##### N-(5-((6,7-Dimethoxyquinolin-4-yl)oxy)-3,4,6-trimethylpyridin-2-yl)-N-(4-fluorophenyl)cyclopropane-1,1-dicarboxamide (**3**)

**16** (120 mg, 0.35 mmol) and *N*,*N*-diisopropylethylamine (73 µL, 0.42 mmol) were added to a mixture of **14** (114 mg, 0.47 mmol) in dichloromethane (4 mL). It was stirred at room temperature for 24 h and diluted with dichloromethane and water. The aqueous layer was extracted with dichloromethane, and the combined organic solution was dried over anhydrous magnesium sulfate, filtered, and concentrated. The residue was purified by silica gel column chromatography (CH_2_Cl_2_:MeOH = 40:1) to give **3** (49 mg, 19%) as a brown solid. R_f_ 0.20 (CH_2_Cl_2_:MeOH = 15:1); m.p. 141 °C; MS *m/z* 545.5 [M + H]^+^; ^1^H NMR (CDCl_3_) δ 10.09 (s, 1H), 8.46 (d, *J* = 5.4 Hz, 1H), 8.25 (s, 1H), 7.62 (s, 1H), 7.52 (dd, *J* = 9.0, 4.9 Hz, 3H), 7.01 (dd, *J* = 11.8, 5.5 Hz, 2H), 6.18 (d, *J* = 5.4 Hz, 1H), 4.09 (d, *J* = 3.6 Hz, 6H), 2.29 (s, 3H), 2.18 (d, *J* = 15.3 Hz, 6H), 1.87 (s, 2H), 1.68 (d, *J* = 2.8 Hz, 2H) (see Appendix A);^13^C NMR (CDCl_3_) δ 167.62(2C), 161.33, 159.35, 157.45, 153.60, 150.14, 147.79, 133.81, 127.74, 122.16(2C), 122.03(2C), 115.73(2C), 115.37(2C), 115.17, 107.08, 101.29, 99.14, 56.34, 56.28, 28.58, 18.76, 18.44, 14.91, 13.21(2C) (see Appendix A); HPLC retention time 15.8 min, purity 96.5% (see Appendix A).

##### N-(5-((6,7-Dimethoxyquinolin-4-yl)oxy)pyridin-2-yl)-N-(4-fluorophenyl)cyclopropane-1,1-dicarboxamide (**4**)

**17** (50 mg, 0.17 mmol) and *N*,*N*-diisopropylethylamine (59 μL, 0.34 mmol) were added to a mixture of **14** (41 mg, 0.17 mmol) in dichloromethane (2 mL). It was stirred at room temperature for 24 h and diluted with dichloromethane and water. The aqueous layer was extracted with dichloromethane, and the combined organic solution was dried over anhydrous magnesium sulfate, filtered, and concentrated. The residue was purified by silica gel column chromatography (CH_2_Cl_2_:MeOH = 20:1) to give **4** (10 mg, 12%) as a beige solid. R_f_ 0.20 (EtOAc:Hex:MeOH = 10:10:1); m.p. 119 °C; MS *m/z* 503.5 [M + H]^+^; ^1^H NMR (CD_3_OD) δ 8.34 (d, *J* = 5.4 Hz, 1H), 8.19–8.14 (m, 2H), 7.62 (dd, *J* = 9.0, 2.9 Hz, 1H), 7.52 (s, 1H), 7.49–7.43 (m, 2H), 7.26 (s, 1H), 6.96 (d, *J* = 8.9 Hz, 2H), 6.46 (d, *J* = 5.4 Hz, 1H), 3.91 (d, *J* = 5.4 Hz, 6H), 1.60 (d, *J* = 2.5 Hz, 2H), 1.57 (d, *J* = 2.6 Hz, 2H) (see Appendix A); ^13^C NMR (CD_3_OD) δ 174.15, 173.20, 164.92, 164.86, 162.50, 157.63, 154.14, 152.94, 152.14, 151.41, 150.12, 144.79,137.84 (d, *J* = 2.9 Hz), 134.97, 127.31 (d, *J* = 8.0 Hz), 119.79, 119.46, 118.88, 118.66, 110.08, 106.76, 103.01, 59.09, 59.05, 33.63, 20.44(2C) (see Appendix A); HPLC retention time 16.5 min, purity 98.8% (see Appendix A).

### 4.2. Biological Evaluation

#### 4.2.1. Cell Lines and Culture

Human cancer cell lines derived from the liver (Hep3B, Huh7) and the lung (A549, H1299) were obtained from American Type Culture Collection (ATCC, Manassas, VA, USA). An H6c7 human normal pancreatic duct epithelial cell line was purchased from Kerafast (Boston, MA, USA). Hep3B, Huh7, and A549 cells were cultured in Dulbecco′s Modified Eagle′s Medium (DMEM) (Hyclone, Logan, UT, USA), and H1299 cells were cultured in an RPMI-1640 medium (Hyclone). The media were supplemented with 10% fetal bovine serum (FBS) (Gibco/ThermoFisher Scientific) and 1% penicillin/streptomycin (Gibco/ThermoFisher Scientific). In the case of H6c7, the cells were maintained in a keratinocyte serum-free medium supplemented with a recombinant endothelial growth factor (rEGF) and bovine pituitary extract (Gibco/ThermoFisher Scientific). All the cells were incubated at 37 °C under a 5% CO_2_ atmosphere.

#### 4.2.2. Proliferation Assay

HCC (Hep3B, Huh7) and NSCLC (A549, H1299) cell lines were seeded in a 96-well plate at a density of 2.5 × 10^4^ cells/mL in complete media and allowed to adhere for 24 h. The cells were serum-starved (1% serum containing medium) for 24 h and were then treated with different concentrations (0.1, 0.3, 1, 3, 10, 30, 100 μM) of cabozantinib (Selleckchem.com, Huston, USA), compound **3**, or compound **4** in the presence of 10% serum. After 48 h, 25 µL of a 3-(4,5-dimethylthiazol-2-yl)-2,5-diphenyltetrazolium bromide (MTT) dye solution (5 mg/mL) (Merck, MA, USA) was added to the wells and further incubated for 4 h at 37 °C. Then, the media were removed, and 200 µL of dimethyl sulfoxide (DMSO) (Duksan, Ansan, South Korea) was added to solubilize the formazan crystal. After a 30 min incubation, the absorbance was measured at 540 nm using a microplate reader (BMG LABTECH, Ortenberg, Germany).

#### 4.2.3. Cytotoxicity Assay

The H6c7 cell line was seeded in a 96-well plate in a complete medium. After 24 h, the culture medium was changed into a keratinocyte serum-free medium. The cells were then incubated with different concentrations (0.1, 0.3, 1, 3, 10, 30, 100 μM) of cabozantinib, compound **3**, or compound **4** for 48 h, and the cell viability was measured using an MTT assay. The optical density was measured at 540 nm using a Spectrostar Nano microplate reader (BMG LABTECH).

#### 4.2.4. *c*-Met Kinase Assay

A *c*-Met kinase assay was performed at the Reaction Biology Corporation (Malvern, PA, USA) using a Kinase HotSpotSM assay platform (www.reactionbiology.com, last accessed at 19 October 2020). Briefly, human *c*-Met kinase (5–10 mU) and peptide substrate (KKKSPGEYVNIEFG, 20 µM) were prepared in a reaction buffer with a final volume of 25 μL. The compounds were delivered into the reaction, followed ~20 min. later by the addition of a mixture of ATP and [γ-33P-ATP] (specific activity approx. 500 cpm/pmol, concentration as required) to a final concentration of 10 μM. After incubation for 40 min. at 25 °C, the reaction was stopped by the addition of a 3% phosphoric acid solution. Then, the reaction was spotted onto a P30 filtermat, and unbound phosphate was removed by washing 3 times for 5 min in 75 mM phosphoric acid and once in methanol prior to drying and scintillation counting. The background counting derived from the control reactions containing the inactive enzyme was subtracted, and specific kinase activity data were expressed as the percentage of remaining kinase activity in the test compounds compared to the vehicle (dimethyl sulfoxide) reactions. IC50 values and curve fits were obtained using GraphPad Prism 5 software.

#### 4.2.5. Protein Extraction and Immunoblotting

The cells (3.5 × 10^5^) were seeded into a 60 mm dish and allowed to attach for 12 h. The cells were treated with cabozantinib (10 μM) or compound **4** (10 μM) for 24 h and 48 h. Cell lysates were collected in a radioimmune precipitation assay (RIPA) buffer containing a protease and phosphatase inhibitor cocktail (Thermo Fisher Scientific). Cell extracts were centrifuged at 17,000× *g* for 15 min, and the protein levels in collected supernatants were measured using a BCA protein assay reagent (Pierce; Rockford, IL, USA). The proteins were separated by SDS-PAGE and transferred at 200 mA to Hybond ECL nitrocellulose membranes (Amersham Life Science, Buckinghamshire, UK) for 1 h. Non-specific binding was blocked using 5% bovine serum albumin (BSA) in Tris-buffered saline (TBS)-Tween 20 (TBS-T) for 1 h. The membranes were then incubated with primary antibodies *p-c*-MET (Y1230/Y1234/Y1235) (ab5662, Abcam, Cambridge, UK), p-c-MET (Y1349) (ab68141, Abcam), c-MET (ab216574, Abcam), and β-actin (sc-8432, Santa Cruz Biotechnology Inc., Dallas, Texas, USA) in 3% BSA for 16 h at 4 °C. The membranes were washed 3 times with TBS-T and incubated with a horseradish peroxidase-conjugated secondary antibody for 1 h at 25 °C. The membranes were then washed, incubated with ECL (Pierce, Appleton, WI, USA) detection reagent, and exposed under a luminescent image analyzer, LAS-4000mini. β-Actin was used as the loading control.

#### 4.2.6. Measurement of Apoptosis

Cells treated with cabozantinib (10 μM) or compound **4** (10 μM) for 48 h were collected and washed with ice-cold PBS twice. Then, the cells were stained with propidium iodide and Annexin V-FITC using a FITC Annexin V apoptosis detection kit (BD Biosciences, San Jose, CA, USA) as per the manufacturer’s instructions. Briefly, the vehicle- or the drug-treated cells (1 × 10^5^) were washed with ice-cold PBS twice and suspended in 1× binding buffer (100 μL). Then, 5 μL of Annexin V-FITC and 5 µL of PI were added, gently vortexed, and incubated for 15 min at 25 °C in the dark. Thereafter, 400 µL of 1× binding buffer was added and analyzed by flow cytometry (FACSVerse Cytometer, BD Biosciences).

#### 4.2.7. Anti-Tumor Activity Measurement in Hep3B Xenografted CAM Tumor Model

Fertilized chicken eggs were purchased from Byeolbichon Farm (Gyeongbuk, South Korea) and incubated at 37 °C with 55% relative humidity. On day 9 of egg incubation, a small hole was made in the shell over the air sac after the selection of bifurcated vessels. Another hole was made on the broadside using a needle by applying negative pressure to the wider part, creating false air sac. Using a grinding wheel (Dremel, Racine, WI, USA), a small window (1 cm^2^) was created in the eggshell above the false air sac. Subsequently, Hep3B human liver cancer cells (1.5 × 10^6^ cells) containing vehicle, cabozantinib, or compound **4** (1 and 5 μM) were suspended in 1:1 ratio of DMEM and Matrigel (Corning Inc., Corning, NY, USA). The mixture was inoculated on the CAM (final drug doses = 30 and 150 pmol/CAM), and the anti-tumor activity of the drugs was compared at the same concentrations. After sealing the window in the eggshell, the eggs were then returned to the incubator. After 4 days of incubation, tumor tissues attached to CAM were resected from the embryo and harvested. The number of vessel branch points within the tumor region was counted and analyzed using ImageJ software. Chick embryo experiments were performed in accordance with the institutional guidelines of the Institute of Laboratory Animal Resources (1996) and Yeungnam University for the care and use of laboratory animals (2009). The experiment protocol was reviewed and approved by the Institutional Animal Care and Use Committee at the Yeungnam University.

#### 4.2.8. Statistics

The statistical significances between groups were determined using a one-way or two-way ANOVA followed by the Student-Newman-Keul comparison method (GraphPad Prism 5.0 software, San Diego, CA, USA). The results are presented as the means ± standard errors of at least three independent experiments. Statistical significance was accepted for *p* values < 0.05.

### 4.3. Molecular Docking

All the registered crystal structures of *c*-Met (https://www.uniprot.org/uniprot/P08581 last accessed at 14 November 2020) were downloaded and aligned to have an identical direction. The separation of 81 receptors and 63 inhibitors in the overlaid structures followed. Shape similarities between the synthesized molecules in this study and the ligands in the complex were calculated using the ROCS module of the Openeye package (Santa Fe, NM, USA) [35]. Omega was prepared the conformational ensemble consisting of 1000 conformers for the small molecules [34]. The metric for quantifying the shape similarity is the TanimotoCombo score, the summed value of shape and color Tanimoto scores. The value lies in the range of 0 to 2, and the closest one to 2 indicates the closest similarity. The conformation showing the highest TanimotoCombo score against the known inhibitors in the crystal structure was chosen as the pose. The structural complementary with the intermolecular contact was visualized and analyzed using the Grapheme Toolkit of Openeye package.

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
