# Peer review of "Novel Pyridine Bioisostere of Cabozantinib as a Potent c-Met Kinase Inhibitor: Synthesis and Anti-Tumor Activity against Hepatocellular Carcinoma"

_ijms, 2021, doi:10.3390/ijms22189685_

Round 1

Reviewer 1 Report

The manuscript "Novel pyridine bioisostere of cabozantinib as a potent c-Met kinase inhibitor: Synthesis and antitumor activity against hepatocellular carcinoma" is a complete an interesting study that could be published in IJMS after revising the following concerns:

1. In certain pages (for example -not limited to-, pages 2, 4, 9, 12), big blank spaces have been left. Please rearrange text or split figures into 2/3 shorter figures to avoid having such big blank spaces.

2. Figure 2 is not readable. Numbers/names of the aminoacids are impossible to read and it is in certain cases difficult to differentiate the type of residue (backbone, side chain or both). Please enlarge the figure, in a way that the right area is removed so that the formula occupies all the page width, to gain resolution. Legend could be embedded in the figure, in one/two corner(s).

3. Compound 3 could be evaluated at least in the proliferation and cytotoxicity evaluation, which is a quick assay that can provide more information. It can be much less potent as c-Met kinase inhibitor, but it still could show cytotoxic activity.

4. Quality of western blot is not excellent. Certain WB have to be repeated for having differential load of the ß-actine.

5. Names of the vendors of the chemicals used in both the synthesis and in the biological assays should be provided.

6. The concentrations at which the compounds were evaluated in cytotoxicity and proliferation assays should be indicated in Material and Methods. Please revise Material and methods to ensure that all conditions like this are indicated in a way that it is possible that an independent researcher can repeat the experiments.

Author Response

Reviewer 1

Comments and Suggestions for Authors

The manuscript "Novel pyridine bioisostere of cabozantinib as a potent c-Met kinase inhibitor: Synthesis and antitumor activity against hepatocellular carcinoma" is a complete an interesting study that could be published in IJMS after revising the following concerns:

  1. In certain pages (for example -not limited to-, pages 2, 4, 9, 12), big blank spaces have been left. Please rearrange text or split figures into 2/3 shorter figures to avoid having such big blank spaces.

--->Thank you for the points. The figures are rearranged, and blank spaces are all removed in the revised manuscript.

  1. Figure 2 is not readable. Numbers/names of the amino acids are impossible to read and it is in certain cases difficult to differentiate the type of residue (backbone, side chain or both). Please enlarge the figure, in a way that the right area is removed so that the formula occupies all the page width, to gain resolution. Legend could be embedded in the figure, in one/two corner(s).

---> We have changed the shape of the Figure 2 according to the reviewer’s suggestion.

  1. Compound 3 could be evaluated at least in the proliferation and cytotoxicity evaluation, which is a quick assay that can provide more information. It can be much less potent as c-Met kinase inhibitor, but it still could show cytotoxic activity.

--->As the reviewer suggested, we performed proliferation and cytotoxicity assay of compound 3, and results are included in Figure 3.

  1. Quality of western blot is not excellent. Certain WB have to be repeated for having differential load of the ß-actine.

--->Three independent sets of WBs had been performed and quantitated by normalization based on ß-Actin. However, we have repeated another set of WBs, and previous ones have been replaced with newly obtained immunoblots.

  1. Names of the vendors of the chemicals used in both the synthesis and in the biological assays should be provided.

--->In the revised manuscript, chemicals in the synthesis were specified in section 4.1.1. The chemical vendors in the biological assays were provided in each experimental section.

  1. The concentrations at which the compounds were evaluated in cytotoxicity and proliferation assays should be indicated in Material and Methods. Please revise Material and methods to ensure that all conditions like this are indicated in a way that it is possible that an independent researcher can repeat the experiments.

--> We have revised the Material and Methods, and the concentrations of chemicals and the detailed conditions are specified.

Reviewer 2 Report

In this manuscript, Karmacharya et al. studied that two novel bioisosteres (compound 3 and 4) of cabozantinib were designed and synthesized by replacement of the centered benzene ring of cabozantinib with trimethylpyridine (for 3) and unsubstituted pyridine (for 4), respectively. In addition, compound 4, which possesses a dual mode of action, inhibition of both activity and expression of c-MET, showed much stronger HCC selectivity and anti-tumor activity than cabozantinib. The results suggest that compound 4 may be one of the promising therapeutic options against HCC. The findings are very interesting. However, some issues need to be addressed before the manuscript is suitable for publication.

Major comment:

  1. Cabozantinib is a multi-kinase inhibitor target (VEGFR2, KIT, RET, AXL...). Therefore, the author should provide compound 4 in vitro kinase assay data, not only focus on c-met inhibition.
  2. What is the effect of compound 4 on tumor cells apoptosis?
  3. What is the toxicity of compound 4 in vivo? Why not perform mice in vivo assay and use CAM tissue assay instead?
  4. How to decide the dose of Figure 5 CAM tissue assay? Furthermore, how to convert it into mice dose?

Minor comment: 

  1. Cabozantinib purchaser from where?
  2. Figure 3, 4, the experimental n value needs to be marked.

Author Response

Reviewer 2

Comments and Suggestions for Authors

In this manuscript, Karmacharya et al. studied that two novel bioisosteres (compound 3 and 4) of cabozantinib were designed and synthesized by replacement of the centered benzene ring of cabozantinib with trimethylpyridine (for 3) and unsubstituted pyridine (for 4), respectively. In addition, compound 4, which possesses a dual mode of action, inhibition of both activity and expression of c-MET, showed much stronger HCC selectivity and anti-tumor activity than cabozantinib. The results suggest that compound 4 may be one of the promising therapeutic options against HCC. The findings are very interesting. However, some issues need to be addressed before the manuscript is suitable for publication.

Major comment:

  1. Cabozantinib is a multi-kinase inhibitor target (VEGFR2, KIT, RET, AXL...). Therefore, the author should provide compound 4 in vitro kinase assay data, not only focus on c-met inhibition.

---> The present study reports the design, synthesis and biological activity of novel bioisosteres of cabozantinib, compound 3 and 4, with a particular focus on c-MET kinase inhibitory activity. As we discussed in the revised manuscript, there was quite good relationship between the c-MET kinase inhibitory activity and anti-proliferative activity of the compounds: Compound 4 which was stronger c-MET inhibitor showed stronger anti-proliferative activity against HCC cell lines than cabozantinib. In the opposite direction, compound having very weak c-MET inhibitory ability had very low anti-proliferative activity against cancer cells. These are the main theme of the present study. Although VEGFR2 kinase inhibitory activity was not determined, the present study examined the effect of compound 4 on VEGFR2 through an indirect method. Tumor-induced angiogenesis reflects the involvement of VEGFR2 signaling. Compared to cabozantinib, compound 4 showed a similar level of inhibitory action against tumor-induced angiogenesis, implicating the degree of VEGFR inhibitory activity of compound 4.

  1. What is the effect of compound 4 on tumor cells apoptosis?

---> According to the reviewer’s suggestion, apoptosis-inducing activity of compound 4 in Hep3B and A549 cells was evaluated by FACS. The apoptosis-inducing activity of compound 4 was greater than cabozantinib. The results are included and explained in new section, 2.5. Comparison of apoptosis-inducing activity of cabozantinib and compound 4.

  1. What is the toxicity of compound 4 in vivo? Why not perform mice in vivo assay and use CAM tissue assay instead?

---> There were no dead embryos among the fertilized eggs treated with compound 4 in the CAM assay.

---> The CAM is a highly vascularized membrane. The CAM model was developed as a model of angiogenesis in the 1970s, and by the 1980s, it was identified as a tool to study tumor metastasis. In the present study, CAM tumor model was chosen over mouse subcutaneous tumor model for the following reasons: 1) tumor-induced angiogenesis in CAM model is easy to quantitate. 2) CAM tumor model is much easier to handle and much cheaper than mouse tumor model.

  1. How to decide the dose of Figure 5 CAM tissue assay? Furthermore, how to convert it into mice dose?

---> Cancer cells were inoculated onto the CAM in a mixture of cell suspension containing drugs and Matrigel (1:1 ratio) to prevent cell scattering. Similarly, drugs (cabozantinib and compound 4) remain in the topical area, thereby, the drug doses are expressed as 30 and 150 pmol/CAM, not per whole egg. If the drug is administered i.v. or i.p., 1 and 5 μM drug concentrations used in the study may be converted to 1 and 5 mg/kg.

Minor comment:

  1. Cabozantinib purchaser from where?

---> Cabozantinib was bought from Selleckchem.com, and it has been specified in the revision.

  1. Figure 3, 4, the experimental n value needs to be marked.

---> In Figure 3, n has been specified as “Results are presented as the mean ± S.E.M. from three independent experiments.” In the Figure 4, n had been already presented as “Bar graphs represent the mean ± S.E.M. from three independent experiments.”

Round 2

Reviewer 1 Report

The manuscript has been significantly improved in respect to the previous version, and the different concerns given have been addressed.

As a small comment, it is possible to see different formats in Materials and Methods (mainly different font size). They should be homogenized so that all the text is in the format of the journal